# Raiders of the Olympic Values: Perception of the Development of Women’s Canoeing in Spain for Tokyo 2021

**DOI:** 10.3390/ijerph19116909

**Published:** 2022-06-05

**Authors:** Juan Carlos Guevara-Pérez, Jorge Rojo-Ramos, Rudemarlyn Urdaneta-Camacho, Emilio Martín Vallespín

**Affiliations:** 1Faculty of Economics and Business, University of Zaragoza, 50005 Zaragoza, Spain; urdanetaruth19@gmail.com (R.U.-C.); emartin@unizar.es (E.M.V.); 2IGOID Research Group, Department of Physical Activity and Sport Sciences, University of Castilla-La Mancha, 45071 Toledo, Spain; 3Motricity and Education (HEME) Research Group, Department of Health, Economy, University of Extremadura, Avda. de la Universidad s/n, 10003 Cáceres, Spain

**Keywords:** women and sport, sports talent, canoeing, Olympic games, gender in sport

## Abstract

Although canoeing is one of the oldest sports in the Olympic program, it was not until the Tokyo Olympics in 2021 that women’s canoeing was first included in the competition. This fact has posed a challenge to the initiation and technification systems of countries in order to obtain competitive results, particularly in Spain, as it is one of the sports that contributed the most medals to the Olympic medal tally. The aim of this study was to evaluate the promotion and development of talent in women’s canoeing in Spain for its first-ever Olympic participation. For this purpose, an analytical survey (*n* = 167) was carried out, the answers to which were contrasted by gender and modality practiced. The results showed a positive evaluation of the current position in flatwater female canoeing regarding talent that is consistent with the competitive results achieved. Additionally, we found that the gender of the respondents influences their perception of the age of sport initiation and the suitability of the progression in the competition systems for the promotion of women’s canoeing in Spain. Therefore, the results of the questionnaire will facilitate a quick diagnosis of critical aspects by sport managers, allowing them to take corrective actions in time for the development of female canoeists and, at the same time, to promote future studies that delve deeper into these topics.

## 1. Introduction

History has been marked by struggles for greater participation of women in different social spheres, of which sport is one of the most visible scenarios. However, it is considered that a culture that marginalizes women could be institutionalized in sports organizations themselves. This is not only because women are underrepresented in sports organizations, but also because, in turn, in terms of gender, the sporting practice seems to adhere to an androcentric model that has been assimilated and institutionalized to the point of being assumed as valid. Within the Olympics, this male hegemony persists in many sports, especially, in canoeing.

Since its incorporation as an exhibition sport at the Paris Games in 1924, canoeing has been present in all subsequent editions and currently counts 16 sporting events, so the number of medals at stake is very significant. Additionally, since Rio 2016, it is also part of the Paralympic program. Even so, after almost a century of Olympic activity, it has not been until the last Olympic Games (JJOO) in Tokyo 2021 that women have been able to participate officially in canoeing for the first time.

In Spain, despite the fact that it is not a sport of the masses, canoeing is one of the disciplines that has contributed the most Olympic medals (19) to Spanish sport achievements [1], and the most successful Spanish athletes at the Olympic Games are canoeists. For many years, Spanish canoeing has enjoyed wide international recognition, hosting events of enormous popularity and with an influential presence in the highest echelons of canoeing at the global level. 

Given the positioning of Spain canoeing in the world, it is pertinent to take a look at the actions developed during the qualifying rounds and participation route of the last Olympic cycle. In addition to the struggle undertaken by countries to obtain a place in the last Olympic Games, Tokyo 2021, the incorporation of women in canoeing has required overcoming a large number of prejudices and stereotypes, given the obstacles involved in the promotion of a modality considered until then masculine. 

In this scenario, the search for effectiveness and efficiency of sports systems is an open topic in the specialized literature, for which the development of talent is a common feature. 

The aim of this study was to evaluate the development of talent in women’s canoeing in Spain for their first Olympic participation. For this purpose, the questionnaire ‘Evaluation of the current position in canoeing with regard to talent ‘of the International Canoe Federation (ICF-Q) was applied and contrasted with the gender and type of boat of the respondents, given the low participation of women and of canoe as a boat within the specialties of canoeing. The study involved 167 individuals (>18 years old) representing 73 clubs and 15 Autonomous Communities (CCAA), who are part of the Spanish canoeing flatwater competitive structure.

The present study is structured as follows: after the introduction, a Section 2 contextualizes the study based on previous literature, from the perspectives of both gender and canoeing. Subsequently, the Section 3 describes the methodology used, whereas the Section 4 shows the results and discusses the findings of the study. Finally, the main conclusions and some future lines of research are presented in the Section 5.

## 2. Previous Literature

In the search for competitive results, the National Governments allocate public resources to sport through the sports federations, which must invest them in the most efficient way in both the promotion and the development of their activity as well as in the preparation of their athletes for international competitions. In Spain, high-level sport is of interest to the National Governments, and the sports federations are responsible for the development of high-performance sport.

In this context, there is a wide literature that attempts to measure the effectiveness of sport systems [2,3,4]. Other studies have focused on the efficiency in the rational use of resources allocated to sport [5,6], and others on the efficacy to lead International Sporting Success [7]. In all these cases, talent generation has proven to be a key aspect to obtain competitive results. All these studies agree on using financial resources and competitive outcomes as the most representative inputs and outputs of the system. However, extensive databases are required to allow robust quantitative analyses when studying efficiency, long time periods to allow qualitative information of scientific value to observe efficacy, and both to observe effectiveness.

The SPLISS project (Sports Policy factors Leading to International Sporting Success) has made important contributions in comparing performance within various organizations and nations, providing theoretical models on how NPSOs (Non-Profit Sports Organizations) can increase their performance through policy formulation and effective analysis of some key performance indicators through nine pillars. The first pillar and only input of the model is represented by the Financial Pillar [7], and the remaining eight pillars (throughputs) identify the support services and systems delivered to athletes, coaches and organizations at each stage of the development process. Therefore, the throughput pillars refer to the optimum way the inputs can be managed to produce the required outputs, i.e., ‘the International Sporting Success’. In this context, Sotiriadou et al. [8] focused on the throughput of Sprint Canoe sport in Australia, observing an overlap between different pillars that makes it difficult to consider them as separate entities [8]. In all these cases, talent generation has proven to be a key aspect of competitive performance.

### 2.1. Gender in Sport

According to Reskin [9], segregation facilitates the reproduction of social inequalities by allowing groups to be defined as different and therefore subject to different reward systems that allow for the illusion of ‘equal’ treatment. We therefore naturally rule out gender segregation in the home, school, university or workplace. However, the premise that sport should be segregated by gender because of average differences in the abilities of women and men has been largely unquestioned [10].

The strong connection of sport with masculinity and the associated lack of women in elite sport is not a new phenomenon [11,12,13]. In this regard, several researchers have highlighted the underrepresentation of women among elite coaches [14,15], in sport refereeing [16], and in decision-making positions in sport organizations [10,17,18]. 

Spanish canoeing reproduces this demographic behavior, with a marked disproportion between men (76%) and women (24%) when averaging the number of participating members in the last two Olympic cycles [19].

Paradoxical though it may seem, when looking at this situation from the perspective of Olympism, we must remember that the founder of the modern Olympic Movement himself, Pierre de Coubertin, once subscribed to a statement that ‘…an Olympiad with females would be impractical, uninteresting, unesthetic and improper…’ [20]. In this respect, we must recognize that there have been advances in the discourse. For example, in the Olympic Charter, a stated facet of the ‘Mission and Role of the IOC’ is ‘to encourage and support the promotion of women in sport at all levels and in all structures with a view to implementing the principle of equality of men and women’ [21]. Additionally, the Brighton Declaration aims ‘to develop a sporting culture that enables and values the full involvement of women in every aspect of sport’ [22]. Finally, the incorporation by the United Nations of the Sustainable Development Goal (SDG) number 9 in the 2030 agenda paves the way for achieving gender equality and empowering women and girls. But ‘what has been preached has not always been carried out’, as in some events of the Olympic program, women’s participation has had to wait a long time. Such is the case of canoeing in one of its Olympic disciplines: canoeing.

### 2.2. Women and Canoeing

Flatwater canoeing is the oldest discipline and the one with the most events in the Olympic program. In Spanish canoeing, it is the most popular discipline and the one that yielded the greatest international results [1]. In this scenario, the incorporation of women’s events in canoeing has challenged the system’s capacity to generate talent to match the competitive success obtained by the rest of the disciplines, which implied obtaining a place at the Tokyo Olympics in 2021 [23]. 

Considering the two types of competition boats in flatwater canoeing, canoe is the less practiced. In this regard, the Winter Championships, due to their large number of participants, have been ideal for studies in canoeing [24]. Observing the participation in the 50th edition of the 2019 Winter Championship (see Table 1) confirmed the ‘double minority’ in the case of women and canoe, which are located in the two segments with the lowest participation in terms of both modality and gender.

This is both because the technique of canoeing is more complicated than that of kayaking [25] and because of the barriers that females must overcome for their incorporation into the practice of canoeing [26].

In this sense, the incorporation of females in canoeing within the program of the Tokyo 2021 Olympic Games has required overcoming the obstacles involved in the promotion of a modality considered until then masculine, having to face a large number of prejudices and stereotypes cultivated during almost a century of male hegemony. 

In Spain, in 2007, the Supreme Council for Sports (Consejo Superior de Deportes or CSD) created the Women and Sport program with the aim of strengthening the economic and logistical support of Spanish female athletes [27]. In addition, under a public–private partnership that since 2016 has incorporated the energy company Iberdrola as the main promoter, the CSD developed the ‘Universo Mujer’ program for the promotion and development of Spanish female sport [28]. All these initiatives created the conditions for the RFEP to make its first call for women canoeists to participate in the Tokyo 2021 Olympic Games in 2018 [29,30].

During this period, women’s canoeing was included for the first time in the pro-gramme of the 2009 World Championship in Dartmouth, Canada, and Spain achieved the first qualification of a woman canoeist in the final (9th position) of the 2017 World Championship held in Racine, Czech Republic [31]. Therefore, the female canoeists, who obtained in 2021 a 5th place in the Olympic Games single competition (C1 500 m.) and gold and silver medals in the world championship of the same year, have set a successful precedent that gives special interest to this study.

## 3. Materials and Methods

### 3.1. Instruments

A 12-question analytical survey was conducted. The first four questions capture the socio-demographic characteristics of the respondents, and the second block of seven items queries perceptions of talent development in women’s canoeing based on the ICF’s ‘Evaluation of the current position in canoeing sport with regard to talent’ questionnaire (ICF-Q) [32]. 

The ICF-Q items match with the SPLISS throughput pillars, except for Pillar 6, Training facilities, as they fall within the governmental competencies of national, regional and local public administrations, and the Pillar 9, Scientific research and innovation, as they fall within the competencies of the university system. Additionally, previous SPLISS studies in canoeing have shown an overlap between pillars 6 and 9 [8]. Table 2 presents the correspondence between the ICF-Q items’ aims and the SPLISS throughput pillars.

The ICF-Q allows, in a very simple way, a perceived first look at key aspects in the production of talent within a sport system, such as the training of coaches, the quality of the initiation and talent recruitment systems, the suitability of competition programs and the synergies within the different levels of sport [23].

Exploratory factor analysis (EFA) was used by the sampling adequacy indices that provided satisfactory results (KMO test = 0.771 and Bartlett test = 478.2; df = 28; *p* = 0.000). Consequently, a factor structure of seven items grouped in one dimension was established. This instrument uses a Likert scale (1–5), with 1 being ‘Strongly disagree’, 2 ‘Disagree’, 3 ‘Neither agree nor disagree’, 4 ‘Agree’, and 5 ‘Strongly agree’. 

Once the EFA was carried out and the structure of the scale was defined, a confirmatory factor analysis (CFA) was conducted to assess the characteristics of the model. For this purpose, the software package AMOS v.26.0.0 (IBM Corporation, Wexford, PA, EE. UU) was applied. The Questionnaire revealed a one-dimensional factor structure composed of seven items, with good and excellent goodness-of-fit values and high reliability (McDonald’s Omega = 0.82).

### 3.2. Participants

The survey was carried out with the collaboration of the Royal Spanish Canoe Federation (RFEP), and 15 of the 17 Spanish Autonomous Communities participated with a representation of 73 clubs. The total sample amounted to 167 participants. The inclusion criteria for this survey were females and men linked to flatwater canoeing (kayakers, coaches, technicians, referees, club managers, etc.) and of legal age (>18 years old).

### 3.3. Procedure

An email was sent, providing information on the purpose of the research, a written informed consent, and the URL to fill the questionnaires. The average time needed to answer the questionnaire was approximately 10 min. Data collection was performed using the Google Forms application, as electronic questionnaires have been proven to save costs and obtain higher participation [33]. The responses obtained were stored in a spreadsheet, facilitating their transformation and statistical analysis. Data collection was carried out between April and May 2020 (before the Olympic Games).

### 3.4. Statistical Analysis

Data were analyzed using the Statistical Package for Social Sciences (SPSS) version 23.0 for MAC (IBM Corporation, Armonk, NY, USA). The Kolmogorov–Smirnov, asymmetry and kurtosis tests were performed to determine whether the data followed a normal distribution. This assumption was not met, so it was decided to use non-parametric tests.

The median scores of the items of the Likert-type questionnaire were obtained from the median value (M_e_) of each item. Data are presented as median and interquartile range (IQR). The Mann–Whitney U test was used to analyze the relationships between the items based on the Likert scale of the ICF questionnaire to evaluate the current position of canoeing with respect to talent [32] according to gender and sport modality, considering that women and canoeing represent two minority segments within the demographic structure of Spanish canoeing.

Cronbach’s Alpha was used to calculate the reliability of each of the dimensions in this study. Following the method of Nunnally and Bernstein [34], reliability values between 0.60 and 0.70 can be considered acceptable, while values between 0.70 and 0.90 can be considered satisfactory.

## 4. Results and Discussion

Of the 167 participants, 62.87% were men, and 37.13% were women. The mean age was 39 years, considering that the youngest participants were 18 years old, and the oldest 68 years old. Table 3 shows the characteristics of the participants.

Table 4 shows the descriptive data and differences by gender and modality for each of the items that make up the ICF questionnaire. For all items, the relationship was positive. Statistical significance was not observed with respect to modality, but it was found with respect to gender for two items. 

Despite the favorable assessment by the interviewees, there are some aspects to highlight. First, although the study showed that coaches have sufficient training to identify and develop athletes for female canoeist in Spain (ICF-Q item 1 = M_e_ ≥ 4), in terms of gender, only 10% of high-level coaches are females [19]. This fact evidences the need for an internal promotion of female coaches [14,15], so there is still a long way to go, even more so, if we consider the importance of the sports coach as a performance factor in Spanish canoeing [35].

Secondly, with respect to the development of women’s canoeing from the initiation stage, the systems implemented by the clubs (item 2), the talent recruitment strategies (item 7), as well as the age at which female athletes start canoeing (item 6), all of them linked to SPLISS throughput Pillar 4 (Talent identification and development system), demonstrate that respondents’ perceptions were similar, showing room for improvement. In this last respect, statistically significant differences were observed according to gender (*p* = 0.09), which invites reflection on the influence of the perception of competitive sport as risky, dangerous and male-dominated at the time of females joining the practice of some sports disciplines [36]. Therefore, promoting a polyvalent and gradual sport initiation is determinant to attracting and retaining sport practice in women [37]. The opposite could contribute to the early abandonment of sport practice by female canoeists, reported in the literature [26].

Thirdly, regarding the competition systems, the study yields a positive assessment with respect to their suitability for the development of female canoeing (item 4) and their contributions to the progression in the development of female canoeing (item 5), both linked to SPLISS throughput Pillar 6 (National competition). In the latter case, a statistically significant relationship was observed as a function of gender (*p* = 0.04), which is relevant when enhancing competitive balance [38]. These results vindicate the systematic exclusion of women from sport competitions for much of the 20th century, which has reinforced the image of sport as a symbolic space to celebrate ‘male virtues’ [39] and the subsequent segregation of women [10]. Finally, the respondents perceived a policy linking Clubs, Autonomous Federations and the RFEP for the development of women’s canoeing in Spain (item 3).

Within the framework of the SPLISS throughputs, pillars 4 (Talent identification and development system) and 6 (National competition) group five of the seven ICF-Q items (see Table 2); in this sense, the statistical significance observed for ICF-Q items 6 (*p* = 0.09) and 5 (*p* = 0.04) highlights the need for a gender intervention on these two pillars, going deeper into the age of initiation of sportswoman (item 6) and addressing the progression of competition programs (item 5), so that they are adjusted to the evolution of the athletes.

Therefore, the present study provides a starting point for studies of efficiency [5,6], efficacy [7], and effectiveness [2,4] of sport systems in different contexts, based on the generation of talent.

One limitation of the present study is the size of the sample, since it comprised adults in a minority sport. The absence of precedents has also been a conditioning factor, since we assessed the perception of the participants with respect to an extraordinary event such as the incorporation of women in an event of the Olympic program. Another limitation of this study is that the items were not translated through a cross-cultural adaptation from English, but rather a literal translation was used to facilitate reading.

On the other hand, one of the contributions of the ICF-Q is to facilitate a quick diagnosis of the most critical topics so that managers can take corrective actions in time and, in turn, focus on qualitative studies such as SPLISS throughputs, particularly, in this case, Pillar 4 (Talent identification and development system) and Pillar 6 (National competition) [8].

In general terms, the study yields a positive assessment of the current position in flatwater canoe women sport with regard to talent, by obtaining scores above the average for all the items of the instrument (M_e_ ≥ 3); this, despite having been carried out before the celebration of the Tokyo 2021 Olympic Games, is consistent with the competitive results obtained. However, the lack of optimal results (5 under Likert scale) suggests that there is still a long way to go. 

In this sense, the current levels of participation have managed to be comparable to those observed in other sports where female competition has a long tradition [27]. This is very commendable in terms of gender, considering that, in Spain, just over one decade ago, it was unusual to see a woman in a competitive canoe.

### Managerial Insights

This section provides some recommendations based on the key observations for sport policy-makers and canoe sport managers. In this regard, the management of sport federations struggles with tensions between high-performance sport and mass participation sport, where the former’s focus on ‘sports development’ does not always align with the latter’s, ‘sport for development’ [40]. 

The criteria established in the RFEP’s project ‘Women’s canoe talent detection for the Tokyo 2020 Olympic Games’ show the search for competitive results in the short term [29]. This fact may compromise the promotion of grassroots sport and coincides with the need to reinforce talent identification and development system policies, for example, at the age of sports initiation, as has been observed in this study (item 6).

In addition, the National competition system requires a review to ensure the sustainability of continuing female participation, as can be seen in the present study (item 5) about the competition development programs. In this respect, some strategic proposals could consider providing clubs with the necessary equipment for initiation and technical training, the training of female coaches in the specialty of canoeing, and a multipurpose initiation (canoe and kayak) that reinforces the sustainability of women’s participation in flatwater canoeing, nationally and internationally.

Finally, the reliability score of the questionnaire was 0.98, which is considered an excellent value, above 0.90, according to Nunnally and Bernstein [34].

## 5. Conclusions

The present study represents a contribution to the scarce literature on canoeing based on an evaluation of the promotion and development of talent in female canoeing in Spain with a view to its first Olympic participation. For this purpose, the ICF-Q was applied to evaluate the current position of canoeing with respect to talent according to gender and sport modality, considering that women and the canoe boat-type represent two minority segments within the demographic structure of Spanish canoeing.

The ICFQ proved to be a fast and versatile tool for the early anticipation of critical areas, thus offering managers of sport organizations the possibility to focus timely corrective actions, as well as to more precisely promote research on efficacy, efficiency and effectiveness of sport systems.

This study provides a positive assessment of the current position in flatwater canoe female sport regarding talent, which corresponds to the competitive results obtained subsequently at both the Tokyo 2021 Olympic Games and the World Championship in the same year. However, in terms of gender equality, there is still a long way to go, since during almost a century of Olympic presence, where canoeing has been exclusively male, an androcentric culture has been cultivated that tends to justify the current inequality.

For all of the above, the promotion of women in canoeing remains a challenge that will require a profound revision of old paradigms that still survive in the collective thinking of some canoeists. 

Future qualitative studies should capture valuable information through in-depth interviews with female canoeists, and other analyses could extend the results by incorporating slalom into the sample and thus provide coverage of both Olympic canoeing modalities. Longitudinal studies that observe the demographic evolution and sport performance through the competitive results of female canoeists during a time series could also be of great value, as well as a comparison of these results with the behavior of the modality at the international level. The generalization of this questionnaire is also an open issue that would allow, through its cross-cultural adaptation to other languages, extending its coverage to other regions for future comparative studies.

It is hoped that this study will contribute to the promotion of female canoeing in Spain and in the world, especially at the beginning of a new Olympic cycle, which usually brings changes in the structures of federations’ governing bodies.

## Figures and Tables

**Table 1 ijerph-19-06909-t001:** Participation canoe-kayak in the ‘L Winter Championship Seville 2019’.

	Gender			Type of Boat		
	Female	Male	Total	Canoe	Kayak	Total
*n*	62	105	167	60	107	167
%	37.13%	62.87%	100%	35.93%	64.07%	100%

Source: Own elaboration based on data collected from the competition history of the Royal Spanish Canoe Federation (RFEP).

**Table 2 ijerph-19-06909-t002:** Correspondences between ICF-Q items’ aims and SPLISS throughputs pillars.

ICF-Q Items’ Aims	SPLISS Throughputs Pillars
(1) Training of coachers	Pilar 7 Coaching provision and coach development
(2) Sport initiation systems	Pilar 4 Talent identification and development system
(6) Age appropriateness for initiation	
(7) Talent Recruitment	
(3) Policy comprehensiveness	Pilar 2 Structure, organization and governance of sport policies: an integrated approach to policy development
	Pilar 3 Foundation and participation
(4) Competition-development programs	Pilar 8 National competition
(5) Competition progression programs	
(8) Athletes’ drop-out rate	Pilar 5 Athletic and post-Career Support

Source: ICF-Q items from Guevara-Pérez et al. [23] and SPLISS throughput pillars from De Bosscher et al. [7].

**Table 3 ijerph-19-06909-t003:** Characteristics of the sample (*n* = 167).

Variable	Categories	*n*	%
Gender	Male	105	62.9
	Female	62	37.1
Age	Under 30	101	60.5
	Between 30 and 40	25	15
	Between 41 and 50	31	18.6
	Over 50	10	5.9
Modality	Kayak	107	64.1
	Canoe	60	35.9
Active in sports	Yes	154	98.2
	No	13	7.8

**Table 4 ijerph-19-06909-t004:** Descriptive analysis and differences by gender and sport modality of the ICF-Q items.

ICF-Q Item		Gender			Type of Boat		
	Total	Women	Men		Kayak	Canoa	
	M_e_ (IQR)	M_e_ (IQR)	M_e_ (IQR)	*p*	M_e_ (IQR)	M_e_ (IQR)	*p*
(1) Do you consider that coaches have sufficient training to identify and develop athletes for women’s canoeing in Spain?	4 (1)	4 (1)	4 (2)	0.25	4 (1)	4 (2)	0.60
(2) Do you consider the current systems for the initiation and development of women’s canoeing in your club to be effective?	3 (2)	3 (2)	3 (2)	0.82	3 (2)	3 (2)	0.52
(3) Do you consider that there is a policy that links the Clubs’ Autonomous Federations and the Royal Spanish Canoe Federation (RFEP) for the development of women’s canoeing in Spain?	3 (1)	3 (0)	3 (1)	0.63	3 (1)	3 (2)	0.79
(4) Do you consider that the current competition programs are adequate to develop women’s canoeing?	3 (1)	4 (1)	3 (1)	0.11	3 (1)	3 (1)	0,80
(5) Do you consider that the structure of the competition program encourages a progression in the development of women’s canoeing?	3 (1)	4 (1)	3 (2)	0.04	3 (1)	3 (1)	0.77
(6) Do you consider the age at which female athletes are currently joining canoeing in Spain to be appropriate?	3 (1)	3 (1)	3 (1.25)	0.09	3 (1.75)	3 (1)	0.21
(7) Do you consider that the current talent recruitment strategies encourage the right type of athletes for the practice of women’s canoeing in Spain?	3 (1)	3 (1)	3 (1)	0.81	3 (1)	3 (1)	0.95
	3.1 (1.1)	3 (1.1)	3 (0.9)	0.27	3 (1.1)	3 (1.1)	0.58

Note: The items were literally translated from English to facilitate reading, without a cross-cultural adaptation to Spanish. M_e_ = median value; IQR = interquartile range. Each score obtained is based on a Likert scale (1–5): 1 means ‘Strongly disagree’, 2 ‘Disagree’, 3 ‘Neither agree nor disagree’, 4 ‘Agree’, and 5 ‘Strongly agree’; *p*: *p*-value from The Mann–Whitney U test.

## Data Availability

The datasets used during the current study are available from the corresponding author on reasonable request.

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
