# Peer review of "Raiders of the Olympic Values: Perception of the Development of Women’s Canoeing in Spain for Tokyo 2021"

_ijerph, 2022, doi:10.3390/ijerph19116909_

Round 1
Reviewer 1 Report
Congratulations on the well-developed research and innovative approach to a specific sport tackling the gender issue
I have only a couple of questions to take into consideration:
- If you have different groups of participants it would be interesting to check if there are differences between them, particularly in the case of the athletes and the rest if you don't have enough samples for further analysis
- It would be important to highlight some limitations of the study and also the most relevant implications, even though there are some proposals for future research but a more specific transfer of knowledge with some specific implementation would be appreciated.
Please, also check line 294 "People The present" to be corrected.
In terms of references, and to increase the number of papers from the 2020s "Spanish women athletes’ performance in the Summer Olympic Games history" by Gonzalez et al. (2020) is missing.
Author Response
Thank you very much for your comments. We appreciate your thoughtful and constructive advice. Below, we try to respond to each of the issues raised in your review. In the new version of the paper, we have incorporated changes to address some of these suggestions and we believe that, as a result, the paper has been significantly improved.

Reviewer 2 Report
Raiders of the Olympic values: perception of the development of women's canoeing in Spain for Tokyo 2021
The aim of this study was to evaluate the promotion and development of talent in women's
canoeing in Spain – specifically in preparation for the first Women Canoe participation in the Olympic.
This is a narrative explanation of the possible reasons for the current and future situations of the sport of canoeing, specifically in women.
Major concerns
- The written English in the manuscript is not ideal with the some of the words used deemed as not appropriate and the sentences’ structure unclear. The manuscript clearly needs to be revised by a native English person.
- Line70-74. I think you need to add some “meat” to this paragraph on the literature review. Need to provide or explain what are the “successful” sport policy in the literature – what are the determining factors of success. Then to try to link this info back to why you feel that the current Spanish canoeing sports policy is similar or different to the mentioned successful sport policy.
- Is the survey questions are in English or Spanish or any other language (since the questions of the survey was taken from the original ICF-Q)? If the survey’s questionnaires are in English, how did the author ensure that all respondents understand the questions fully. However, if the survey in Spanish but was translated from another language, say English or French or German in its original form, how did the authors ensure that meaning of the question is not lost in the translation? Is there any support to suggest that the translation has been verified and validated by some preliminary investigation?
Minor issues:
- the term “modalities” to define the 2 different sports of canoeing and kayaking. I feel other terms such as “specialization” or “boat-mode” are better terms.
- line 52-53. “…promotion of canoe within the modalities of canoeing”. One awkward sentence here.
- Line 55-56. “… any of the Spanish canoeing levels” Should it be Spanish canoeing competitive structure or national competition levels?
- line 57. Define the document. Unclear whether you are referring to the present article or is it the survey that you are doing in the present study?
- Line 59. What is “particularities of canoeing”?
- Line 57 to 62. These sentences can be condensed into a box figure or flow chart – for ease of clarity to readers
- line 64. What is “States”?
- line 88. “licenses? Be more specific “canoe qualifiers”?
- Line 109-110. “In Spain, it is the most popular discipline and where the sport has obtained the greatest international results.” You need evidence of fact and/or other form of support for this statement.
- line 130. Define “behavior”. Unclear term here.
- line136-139. Who has had obtained 5th place? Is it the country Spain or a person?
- Line 147. The word “allows” was used twice in the same sentence. Please change.
- Line 156. What is “CFA”. First time reader is seeing this abbreviation.
- line 157-158. This is in Spanish and hence does not contribute to the article.
- line 180. Delete the “or not”.
- line 256-257. Need to define what you mean by “seem to be appropriate”.
- line 290. What would the authors suggest to be the “optimal results” in the opposite case.
Author Response

(The authors gave the same response as above.)

Reviewer 3 Report
This paper is about the development of women canoe in Spain. In 2021 for the first time a women’s contest has been included in the olympic games. With a survey, the development and promotion of canoeing in Spain was assessed.
In general the topic is relevant and interesting. But the presented data in the tables and the text is some times doubled and probably sometimes mixed up. This makes a complete overview difficult. Based on the presented data it is difficult to follow the conclusion.
What were the actions of the canoe federation on order to establish a female team for the olympics?
In the conclusion, I would recommend to make statements based on our findings and probably give advice how the situation can be ameliorated.
General comments.
Please avoid presenting the same data in the text and in the table. The result section should therefore be partially rewritten.
Specific comments. The last sentence of the abstract is weak. What is the main outcome or respectively a follow up of your study to promote and establish women in sport?
A lot of information is about Spain. I have no doubts that the content is true, but clear references are missing very often. Is it possible to reference some of the information proceed, based on information and content as example of the national olympic federation?
L.118. please do not include the same data in the text and in table 1. Regarding the gender there might be a mix up between male and female. Are 63% men (text) or female (table1)?
Maybe your point would be clearer if you include the numbers for male and female for the two disciplines.
L133. are there other mayor events like world championships were it was possible so fare to compete on an international level as a women? If yes what was the level of the Spanish women team?
Is it true that at the winter championship 167 athletes were competing and 167 other participants were included in the survey?
The group of participants including athletes (m,f) coaches, technicians, referees, club manager seems to the rather heterogenous. Would it be possible to make different probably more homogenous groups?
Table 3. Should be mean values and the IQR be presented with more digits?
I like the discussion about the competition system but I feel the differences should be more presented in the introduction so that he reader can understand your valuable comments.
L294. please check sentence. What people?
Compared to other sports, is the female situation in canoeing different? Are the values that you have presented here an agreement with other sports in Spain? How they compare to sports were female competing has a longer tradition?
Author Response

(The authors gave the same response as above.)

Round 2
Reviewer 2 Report
None
Author Response
Dear Reviewer,
Thank you very much for your receptivity to our article.
Best regards.
Reviewer 3 Report
Thank you for the revision. The manuscript has improved substantially.
Author Response

(The authors gave the same response as above.)
